# CRISPR/Cas9-Mediated Knock-Out of dUTPase in Mice Leads to Early Embryonic Lethality

**DOI:** 10.3390/biom9040136

**Published:** 2019-04-04

**Authors:** Hajnalka Laura Pálinkás, Gergely Attila Rácz, Zoltán Gál, Orsolya Ivett Hoffmann, Gergely Tihanyi, Gergely Róna, Elen Gócza, László Hiripi, Beáta G. Vértessy

**Affiliations:** 1Institute of Enzymology, RCNS, Hungarian Academy of Sciences, H-1117 Budapest, Hungary; racz.gergely@ttk.mta.hu (G.A.R.); tihanyi@stud.uni-heidelberg.de (G.T.); 2Doctoral School of Multidisciplinary Medical Science, University of Szeged, H-6720 Szeged, Hungary; 3Department of Applied Biotechnology and Food Sciences, Budapest University of Technology and Economics, H-1111 Budapest, Hungary; Gergely.Rona@nyulangone.org; 4Department of Animal Biotechnology, Agricultural Biotechnology Institute, National Agricultural Research and Innovation Centre, H-2100 Gödöllő, Hungary; zoltan.gal89@gmail.com (Z.G.); hoffmannorsi@gmail.com (O.I.H.); 5Department of Biochemistry and Molecular Pharmacology, New York University School of Medicine, New York, NY 10016, USA; 6Perlmutter Cancer Center, New York University School of Medicine, New York, NY 10016, USA

**Keywords:** dUTPase, CRISPR/Cas9-mediated knock-out, blastocyst outgrowth, embryonic development

## Abstract

Sanitization of nucleotide pools is essential for genome maintenance. Deoxyuridine 5′-triphosphate nucleotidohydrolase (dUTPase) is a key enzyme in this pathway since it catalyzes the cleavage of 2′-deoxyuridine 5′-triphosphate (dUTP) into 2′-deoxyuridine 5′-monophosphate (dUMP) and inorganic pyrophosphate. Through its action dUTPase efficiently prevents uracil misincorporation into DNA and at the same time provides dUMP, the substrate for de novo thymidylate biosynthesis. Despite its physiological significance, knock-out models of dUTPase have not yet been investigated in mammals, but only in unicellular organisms, such as bacteria and yeast. Here we generate CRISPR/Cas9-mediated dUTPase knock-out in mice. We find that heterozygous *dut* +/– animals are viable while having decreased dUTPase levels. Importantly, we show that dUTPase is essential for embryonic development since early *dut* −/− embryos reach the blastocyst stage, however, they die shortly after implantation. Analysis of pre-implantation embryos indicates perturbed growth of both inner cell mass (ICM) and trophectoderm (TE). We conclude that dUTPase is indispensable for post-implantation development in mice.

## 1. Introduction

The maintenance of genome integrity and faithful preservation of genomic information are crucial for viability. Toward these goals, various DNA damage and repair pathways along with the regulation of a well-balanced deoxynucleotide (dNTP) pool work hand in hand [1]. Nucleotide pools are maintained by several families of dNTP hydrolyzing enzymes present in most organisms [2,3,4,5]. These enzymes sanitize the nucleotide pool by removing nucleotide building blocks (dNTPs) that contain erroneous bases from the polymerase action. The deoxyuridine 5′-triphosphate nucleotidohydrolase (dUTPase) family of enzymes is responsible for the removal of dUTP from the nucleotide pool by hydrolyzing it into dUMP and inorganic pyrophosphate [6,7,8]. The importance of this enzymatic action is evident in light of the fact that most DNA polymerases cannot distinguish 2′-deoxyuridine 5′-triphosphate (dUTP) and 2′-deoxythymidine 5′-triphosphate (dTTP) and will readily incorporate the uracil analog if it is available in the cellular dNTP pool [9]. Through their enzymatic action that generates 2′-deoxyuridine 5′-monophosphate (dUMP), dUTPases also feed into the de novo thymidylate biosynthesis pathway by providing dUMP as the substrate for thymidylate synthase. Two major families of dUTPases have evolved that are referred to as trimeric and dimeric dUTPases, reflecting their corresponding quaternary structure [6,10,11,12,13].

Trimeric dUTPases are present in almost all free-living organisms with the notable exception of trypanosomes. Subunits of these enzymes contain a beta-sheeted arrangement [6]. The three subunits donate conserved sequence motifs to build the three active site of the dUTPase trimer [14,15,16]. This family of dUTPases is characteristic for Archaea, Bacteria, and Eucarya. As such, mammalian species also rely on the action of trimeric dUTPases to keep a well-balanced dUTP/dTTP ratio. Herpesviruses encode an intriguing monomeric homolog of this dUTPase enzyme family [17], where the protein sequence contains a species-specific insert to allow the construction of the usual beta-sheeted dUTPase fold in a monomeric enzyme [18,19]. Dimeric dUTPases perform the same catalytic action, however, the protein sequence and the alpha-helical protein fold are drastically different from those observed in the trimeric dUTPase family [10,11].

Due to the highly significant enzymatic character of dUTPases, the essentiality of this enzyme family was addressed in numerous different organisms. Knock-outs have been generated in several bacteria: *Escherichia coli* [20] and *Mycobacterium smegmatis* [7]. Based on these studies it was argued that the bacterial cells lacking dUTPase activity are not viable. However, genomic analysis of Archaea and prokaryotes identified several species that lack *dut*, the dUTPase encoding gene [12]. These findings indicate that the presence of dUTPases may not be a universal requirement in prokaryotes.

The physiological role and importance of dUTPase have also been addressed in eukaryotes. In yeast, dUTPase knock-out was still viable, although this genotype led to a thymine auxotroph phenotype [21]. In *Caenorhabditis elegans*, RNA-silencing studies indicated that dUTPase might be important in embryonic development [22]. Very recently, in planarians, silencing of *dut* caused lethality in adult animals possibly due to genomic DNA fragmentation. Co-administration of the thymidylate synthase inhibitor 5-fluoro-uracil (5-FU) resulted in more DNA breaks and earlier planarian death [23]. In *Drosophila melanogaster*, dUTPase silencing led to early pupal lethality suggesting a specific role of dUTPase and uracil-DNA metabolism in metamorphosing insects [24,25]. It has been shown that dUTPase is also essential in *Arabidopsis thaliana.* In these plants, reduced dUTPase activity caused DNA damage and increased homologous recombination events. Furthermore, these plants were extremely sensitive to 5-FU [26]. In human cell lines, several laboratories published siRNA dUTPase-silencing studies [27,28,29]. These all proposed that highly efficient silencing with practically no remaining dUTPase does not perturb the cellular phenotype under normal conditions [29]. Still, the dUTPase-silenced cell lines showed increased sensitivity towards inhibitors of the de novo thymidylate biosynthesis. These findings corroborated the clinical significance of dUTPase inhibition in anticancer chemotherapies [30,31,32]. To our best knowledge, knock-out studies on dUTPases have not yet been published for any mammalian species.

Motivated by the lack of knowledge in the field, we initiated dUTPase knock-out experiments in mice. Here we report successful generation of dUTPase knock-out mice using CRISPR/Cas9-mediated genome editing. We find that absence of dUTPase leads to early embryonic lethality. No homozygous knock-out offspring could be observed, however, homozygous knock-out blastocysts are still viable and can be cultured in vitro, suggesting that lethality of the dUTPase knock-out sets in around or shortly after implantation.

## 2. Materials and Methods

### 2.1. CRISPR Constructs

The T7 single-guide RNA (sgRNA) harboring the protospacer-adjacent motif (PAM) sequence and the Cas9 mRNA were obtained from Sigma-Aldrich (St. Louis, MO, USA). According to our request, the designed sgRNA targeted the first common exon of dUTPase isoforms (exon 2) on mouse chromosome 2 (Figure 1a).

### 2.2. CRISPR/Cas9 Efficiency Test in Mouse Embryonic Fibroblast (MEF) Cells

Target sgRNA and Cas9 nuclease mRNA were transfected into mouse embryonic fibroblast (MEF) cells by Lipofectamine™ 3000 Transfection Reagent (ThermoFisher Scientific, Waltham, MA, USA). According to the manufacturer’s recommendation, 2.5 μg Cas9 mRNA and 250 ng target sgRNA were added to the sub-confluent cultures of cells grown in six-well plates. 24 h after transfection, cells were maintained in a fresh medium for 24 h, and then the genomic DNA was extracted with a MasterPure™ DNA Purification Kit (Epicentre, Madison, WI, USA). After DNA amplification with Cel-1-F and Cel-1-R primers, Cel 1 cleavage assay was performed using the Transgenomic^®^ SURVEYOR^®^ Mutation Detection Kit according to the manufacturer’s instructions. The primers used in this study were synthesized by Sigma-Aldrich and are listed in Appendix A.

### 2.3. Animals

dUTPase wild-type and heterozygous mice used in the experiments were produced and maintained in the Animal Care Facility at the Agricultural Biotechnology Institute, National Agricultural Research and Innovation Centre (NAIK) (FVB/N background, Envigo, UK). Animals were housed in groups of 2–5 with free access to food and water. Animals were kept under a standard light–dark cycle (06.00–18.00 h) at 22 °C. This study was carried out in strict accordance with the recommendations and rules in the Hungarian Code of Practice for the Care and Use of Animals for Scientific Purposes. The protocol was approved by the Animal Care and Ethics Committee of the Agricultural Biotechnology Institute, NAIK and the Pest County’s governmental office (permission number: PEI/001/329-4/2013). The method used for euthanasia was cervical dislocation. All efforts were made to minimize suffering.

### 2.4. Micromanipulation and Detection of Gene Targeting

Microinjection was performed as described previously [33]. Briefly, mouse zygotes were collected at 20 h after injection of human chorionic gonadotropin (hCG) from superovulated FVB/N females mated with FVB/N males. Pronuclei were injected using a manual injector with continuous flow. Following visualization of pronuclear swelling, the needle was pulled out through the cytoplasm, injecting a small amount of additional RNA delivery to the cytoplasm. The microinjection mix contained a sgRNA (Sigma-Aldrich, St. Louis, MO, USA) diluted in 10 mM Tris, 0.2 mM EDTA (pH = 7.4) in a final concentration of 10 ng/µL and Cas9 mRNA (Trilink, San Diego, CA, USA) in 10 mM Tris, 0.2 mM EDTA (pH = 7.4) in a final concentration of 150 ng/µL. Microinjections were finished within 2 h after zygote isolation. Injected zygotes were transferred to pseudopregnant CD-1 females (Envigo, Huntingdon, UK). All animals born from embryo transfer were genotyped by polymerase chain reaction (PCR) and T7 assay.

### 2.5. Cloning and Sequencing

Genotyping of the heterozygous founder animals was carried out by amplifying the CRISPR target sites from genomic DNA using primers pBS-F and pBS-R (listed in Appendix A), and the fragments were cloned into *Sal*I/*Eco*RI sites of vector pBluescript SK (+) (Stratagene). Twenty individual bacterial colonies were purified with NucleoSpin^®^ Plasmid DNA Purification Kit (MACHEREY-NAGEL GmbH & Co. KG (Düren, Germany), according to the manufacturer’s instructions, then DNA samples were subjected to sequencing. Based on the sequencing results, two animals (founder #2 and #4) showed CRISPR events, their offspring were termed (D6, M1) and D47, respectively. All DNA samples in this study were verified by sequencing by Microsynth Seqlab GmbH.

### 2.6. Off-Target Analysis

Off-target effects of CRISPR/Cas9 nucleases were evaluated via the online predictor CCTop—CRISPR/Cas9 target (https://crispr.cos.uni-heidelberg.de/) [34]. Twelve candidate loci for the target site with high potential cleavage in the mouse genome were chosen. The selected potential off-target sites were PCR-amplified using genomic DNA from both wild-type mice and the founder animal #4 and evaluated by DNA sequencing. Ten of the twelve candidate off-target sites could be analyzed. In all these cases no difference was observed (see Appendix A). Two of the candidate off-target sites (Off-2 and Off-4) could not be evaluated due to non-specific PCR product from repetitive elements in both the wild-type and the founder #4 samples. The information on the off-target loci identified by the online program are shown in Table 1 and sequencing primer pairs used are listed in Appendix A.

### 2.7. Genotyping

The genotypes of mice were determined by PCR of the total genomic DNA extracted from mouse tails. Genotyping of embryos was also performed by PCR either by isolating DNA from full embryos or from outgrowth assays. All isolated samples were dissolved in a DNA lysis buffer (0.1 M Tris-HCl, pH = 7.4, 0.2 M NaCl, 5 mM EDTA, 0.2% SDS) and DNA was extracted with phenol–chloroform. MyTaq polymerase (Bioline) was activated at 95 °C for 5 min, and PCR was performed for 30 cycles at 95 °C for 30 s, 64 °C for 30 s, and 72 °C for 30 s, with a final extension at 72 °C for 10 min using primers Dut-gen-F and Dut-gen-R (Appendix A). Genotyping from blastocysts was performed by semi-nested PCR using 1 µL template from 30× diluted primary PCR product with primers Dut-nest-F and Dut-gen-R (Appendix A) under the same reaction conditions. DNA fragments were visualized by 1% agarose gel electrophoresis.

### 2.8. Analysis of Dissected Embryos

For the analysis presented in this manuscript, D47 heterozygous males were mated with D47 heterozygous females, and embryos at various stages (3.5–9.5 dpc (days post coitum)) were collected from pregnant D47 heterozygous females. Dissections were performed in ice-cold phosphate buffered saline (PBS). After dissection, embryos were examined and photographed with a LeicaM205FCA-FC fluorescent stereo microscope linked to a DFC7000-T Leica camera. The day of plug formation was defined as embryonic day 0.5.

### 2.9. Analysis of Blastocyst Outgrowth

Embryos were flushed out from the uteri of pregnant mice at 3.5 dpc in M2 medium. Blastocysts were individually cultured on 0.1% gelatin-coated, 12-well tissue culture dishes (Eppendorf), in KO-DMEM ES cell culture medium supplemented with 1000 U/mL LIF and 20% fetal bovine serum (HyClone), in 5% CO_2_ at 37 °C for four days. Outgrowths were photographed daily using a LeicaM205FCA-FC fluorescent stereo microscope linked to a DFC7000-T Leica camera. On the fourth day of culture, outgrowths were photographed, subsequently removed, and genotyped by PCR as described above.

### 2.10. Western Blot

Embryos at 10.5 dpc were dissected immediately following euthanasia of pregnant mice, then washed with PBS and resuspended in a lysis buffer (20 mM HEPES, pH = 7.5, 420 mM NaCl, 1 mM EDTA, 2 mM dithiothreitol (DTT), 25% glycerol). Homogenization was assisted with vortex until the tissue was sufficiently disrupted. Samples were centrifuged at 20,000× *g* for 15 min at 4 °C to remove the insoluble fraction, then the supernatant samples were boiled with SDS buffer at 95 °C for 5 min. Total proteins were resolved under denaturing and reducing conditions on a 12% polyacrylamide gel and transferred to the PVDF membrane (Immobilon-P, Merck Millipore, Billerica, MA, USA). Membranes were blocked with 5% non-fat dried milk in TBS-T (25 mM Tris-HCl, pH = 7.4, 140 mM NaCl, 3 mM KCl, 0.05% Tween-20) for 1 h at 4 °C and were developed against dUTPase (1:2000, Sigma-Aldrich) and α-actin (1:1000, Sigma-Aldrich) for loading control. After applying horseradish peroxidase-coupled secondary antibodies (Amersham Pharmacia Biotech, Piscataway, NJ, USA), immunoreactive bands were visualized by an enhanced chemiluminescence reagent (Millipore, Western Chemiluminescent HRP substrate), and images were captured by a Bio-Rad ChemiDoc™ MP Imaging System. Densitometry was done using Bio-Rad Image Lab™ 6.0 (Hercules, CA, USA).

### 2.11. Statistical Analysis

Statistical analysis was done with a two-sample, single-tailed t-test assuming equal variance using Microsoft Excel or was carried out with a two-sided Mann–Whitney U test. The data were considered significant when *p* < 0.05 (*).

## 3. Results

### 3.1. Targeted Knock-Out of Mouse dUTPase by CRISPR/Cas9 Gene Editing

To explore the importance of dUTPase in mammalian life, we attempted to establish dUTPase knock-out mice using CRISPR/Cas9-mediated gene editing. Figure 1 shows the outline of the applied knock-out strategy. The respective single-guide RNA was designed to disrupt both nuclear and mitochondrial isoforms of dUTPase (Figure 1a), which arise from a single gene (*dut*) on chromosome 2. Efficiency of the CRISPR-mediated events were first assessed by the surveyor assay in mouse embryonic fibroblast (MEF) cells (Figure 1b). We found evidence of CRISPR/Cas9-induced cleaved products and, based on this result, we started the mouse zygote microinjection. Figure 1c shows the schematics of gene targeting and mice generation. In the microinjection experiments, 107 embryos were flushed and microinjected, then 76 embryos from these 107 embryos were transferred to five foster mothers. Two foster mothers had in sum 15 offspring (resulting in a 20% survival rate). Three out of the 15 offspring were successfully targeted by CRISPR/Cas9 based on T7 assay (20% targeted rate), and two of them (these are numbered as #2 and #4) showed mono-allelic indels verified by sequencing. Sequencing results from mouse #2 showed a six bp deletion and one base substitution leading to altered amino acid sequences. Sequencing results from mouse #4 showed a 47 bp deletion leading to frameshift and early stop codons (Figure 1d). Chromatograms of the sequencing could be found in Appendix A. Heterozygous mouse #2 and mouse #4 were the founder (F0) animals for the downstream mouse strains. We have termed the strain from the founder mouse #2 as (D6, M1) and the strain from the founder mouse #4 as D47. Genotype was determined by PCR throughout the study (see ‘Materials and Methods’ section for details). Founder mouse #2 and chimeric founder mouse #4 were then bred to a wild-type mouse for germline transmission thus generating progeny containing the targeted gene. Homozygous and heterozygous offspring of strain (D6, M1), as well as heterozygous offspring of strain D47 showed no gross visually observable abnormalities and were fertile through multiple generations. Further experiments presented in this study were conducted on F2 or even later generations of the mouse strain D47, thereby excluding the possibility of mosaicism.

In our CRISPR-based knock-out experiments, the target site was situated within the *dut* gene, as shown in Figure 1a. The sequence of this target site was as follows: CGCGCGCGGACCCGCGGGT. To check for potential off-target effects, we had carefully analyzed the twenty most similar sequences to this target sequence within the mouse genome with the online predictor software CCTop [34] (see ‘Materials and Methods’ section for details) (Table 1). In each similar sequence, at least three or more mismatches occurred reducing the chance of off-target effects [35]. To decide whether any of the first ten sequences may lead to off-target cleavage using the CRISPR/Cas9 method, we sequenced these genomic segments both in the wild-type and in the founder mouse #4. We also sequenced the 16th and 18th most similar genomic segments as they were located on the same chromosome (chr2) as the *dut* gene, thus, an off-target effect in these segments could be inherited together with the *dut* knock-out allele. Sequences from the wild-type and the mouse #4 were identical (see Appendix A for the sequencing chromatograms). However, two of the candidates’ off-target sites (Off-2 and Off-4) could not be evaluated due to non-specific PCR product in both the wild-type and the founder #4 samples. We therefore concluded that no CRISPR-induced event could be observed at these successfully sequenced sites, arguing for lack of off-target effects.

### 3.2. Analysis of Developmental Effects of dUTPase Knock-Out

To assess the genotype of the embryos and progeny from *dut* +/– mouse intercross breeding we designed appropriate primers for genotyping PCR reactions. The resulting products were different in length from the wild-type as compared to the knock-out allele in mouse strain D47 (as visualized in Figure 2a). This semi-nested PCR method proved to be efficient for genotyping from low cell number containing samples, like blastocysts. *Dut* +/+ and +/– embryos could be detected at 9.5 dpc (Figure 2b) and were obtained by intercrossing D47 heterozygous mice. However, *dut* −/− embryos could only be observed at early pre-implantation stages. We revealed that all three genotypes (+/+, +/–, −/−) resulted in live 3.5-day-old embryos (Figure 2b).

Table 2 summarizes the results of the genotype analysis of the animals investigated in this study. The data clearly showed that *dut* −/− knock-out cells could only be isolated in early embryonic development at the blastocyst stage (3.5 dpc).

### 3.3. Embryonic Development in the dUTPase Knock-Out as Compared to the Heterozygous and Wild-Type Animals

To further assess early embryonic development of dUTPase knock-out embryos, we cultured in vitro for several days of 3.5 dpc blastocysts derived from intercrosses of *dut* +/– mice. After hatching, these blastocysts were attached to a gelatin-coated surface (Figure 3). Blastocyst development was checked after one day and four days (i.e., at 4.5 and at 7.5 dpc). Attached blastocysts formed an inner cell mass (ICM) outgrowth around which trophoblast giant cells were visible (indicated with arrows in Figure 3a). It is shown in these pictures that the *dut* −/− ICM was smaller, while we did not observe obvious changes in the heterozygotes as compared to the wild-type. Additional phase contrast images of embryos can be found in Appendix A. Quantitative analysis of ICM and trophectoderm (TE) regions were performed as described in the Appendix A. Further development of the attached embryos revealed that at 7.5 dpc, the ICM clump size was significantly smaller in the *dut* −/− embryo as compared to the heterozygous (*p* = 0.045) and the wild-type (*p* = 0.022) embryos, while ICM of the heterozygotes did not show significant difference from the wild-type (*p* = 0.14) (Figure 3b). Additionally, the TE size was significantly reduced in the *dut* −/− embryos as compared to the heterozygous embryos (*p* = 0.015) (Figure 3c). Comparison of the wild-type and homozygous embryos showed a considerable reduction of the trophectoderm of the homozygous embryos (*p* = 0.060), while the trophectoderm size of wild-type and heterozygous embryos did not differ significantly (*p* = 0.27). These findings indicated that dUTPase deficiency impairs outgrowth of both ICM and trophectoderm cells.

Next, embryos were isolated at 8.5 dpc or at 9.5 dpc for investigations at post-implantation embryonic stages (Figure 4). No obvious changes could be seen in the development of heterozygotes (Figure 4b,d) as compared to the wild-type (Figure 4c,e). We also observed resorbed embryos (Figure 4a, counted in Table 2) where the genotype could not be evaluated unequivocally since the mother’s decidual tissue could not be separated from the resorptions. No *dut* −/− embryos were found at these or further stages. More images of wild-type, heterozygote, and the resorbed embryos at 8.5 dpc or at 9.5 dpc are provided in Appendix A.

To analyze dUTPase protein levels in the wild-type and heterozygous mouse, Western blot analysis was performed on the total protein extract from *dut +/+* and +/– embryos (Figure 5). dUTPase levels were significantly reduced in the heterozygote as compared to the wild-type (*p* = 0.019). This finding confirmed that the *dut* gene was successfully disrupted using the CRISPR/Cas9 genome editing system. *Dut* −/− homozygous embryo could not be found at this embryonic stage (at 10.5 dpc).

## 4. Discussion

The schematic pattern of embryonic development in mice is illustrated in Figure 6. In this scheme, the timeline of maternal mRNA and protein degradation and the parallel zygotic genome activation are also indicated [36,37]. Our results showed that the dUTPase knock-out did not affect the first several duplication cycles and viable blastocysts could be isolated. Furthermore, isolated *dut* −/− blastocysts could grow further in in vitro cultures, although outgrowth of both the inner cell mass and the trophectoderm cells were impaired. However, further development following implantation was prevented in the homozygous knock-out embryo.

We hypothesize that the lethality of the dUTPase knock-out in mice is probably related to the lack of its enzymatic function in the *dut −/−* embryo. It is well known from the literature that the dUTPase enzyme is a key protein in genome integrity, and its deficiency in bacterial, yeast, and *Drosophila* models led to increased DNA damage frequency [7,20,21,24,25]. Elevated levels of ssDNA and double stranded breaks were observed in these models. The underlying mechanism of the increased DNA damage in lack of dUTPase might be attributed to the potential expansion of the cellular dUTP pool that leads to increased incorporation of uracil moieties into DNA. Numerous uracil moieties within the genomic DNA may induce a hyperactive base excision process through the uracil-DNA glycosylase enzymes [6]. Repair synthesis also occurs in the dUTP-enriched milieu therefore leading to further genomic uracil enrichment. These circumstances transform the repair process into a hyperactive futile cycle.

Our results also showed that at the early stages of embryonic development, until the blastocyst stage, embryos with the *dut −/−* genotype were still viable. Generally, maternal storages are already depleted at the blastocyst stage, however, the exact situation is not yet characterized for maternal dUTPase mRNA, and protein stores. We therefore conclude that viability of the early embryonal stages might be due to various reasons: i) Maternal source of dUTPase may be sufficiently present in these embryonal stages, or ii) the repair mechanism relying on uracil-DNA glycosylase may not yet be effective in these stages. Unfortunately, these aspects are also not yet studied in the literature, necessitating further detailed molecular investigations.

We conclude that mitotic events may proceed in lack of dUTPase in mice, however, the enzyme is indispensable for later differentiation stages. Our model will be useful in later detailed studies to outline the molecular events leading to the observed early embryonic lethal phenotype. In addition, combined multiple knock-outs of *dut* and other relevant genes are also expected to provide important insights into developmental processes.

## Figures and Tables

**Figure 1 biomolecules-09-00136-f001:**
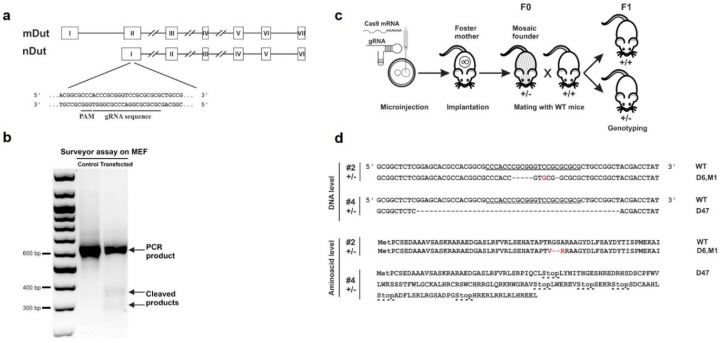
Generation and assessment of CRISPR knock-out mice. (**a**) Schematic diagram of the *dut* gene encoding the nuclear (nDut) and mitochondrial (mDut) isoforms of deoxyuridine 5′-triphosphate nucleotidohydrolase (dUTPase). Exons are indicated with Roman numerals in rectangles, introns are simplified as lines (for longer introns lines are broken). Guide RNA (gRNA) target site and protospacer-adjacent motif (PAM) sequence in the first common exon of the two isoforms are underlined. (**b**) Surveyor assay performed on mouse embryonic fibroblast (MEF) cells used for the detection of indel events induced by transfection with CRISPR gRNA and Cas9 mRNA. The two lower fragments indicate cleavage of the DNA due to CRISPR activity. These are lacking in the control while they are visible in the transfected sample. (**c**) Schematic diagram showing the generation of CRISPR-targeted knock-out mice. Fertilized oocytes microinjected with gRNA and Cas9 mRNA were implanted into foster mothers. The resulting founders (F0) #2 and #4 were cross-bred with wild-type (WT) mice to generate wild-type (*dut* +/+) and heterozygous (*dut* +/–) offspring (F1) containing the targeted locus through germline transmission. (**d**) DNA and predicted amino acid sequence of the two heterozygous founder mice (#2 and #4) showing CRISPR events, compared to the WT. Mouse #2 showed deletion of six nucleotides and a C to G mutation (D6, M1) resulting in the deletion of two amino acids and change of another two. In mouse #4, 47 nucleotides were deleted (D47) which resulted in a frameshift mutation leading to early stop codons indicated with dashed lines. CRISPR target site including PAM sequence is underlined.

**Figure 2 biomolecules-09-00136-f002:**
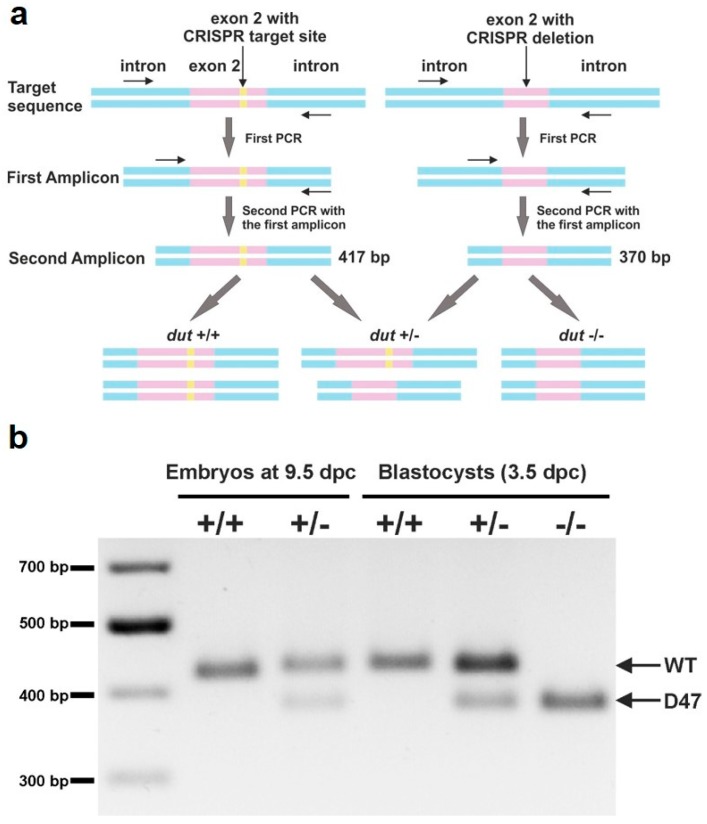
Genotyping of blastocysts. (**a**) Schematic representation of the used semi-nested design for genotyping. Introns are shown in blue, exons are shown in pink, and the CRISPR target site is shown in yellow. DNA isolated from blastocysts was subjected to PCR with primers (shown as arrows) adjacent to the CRISPR target site. The resulting amplicon was used in a second round of PCR with the same reverse and a nested inner forward primer to generate a 417 bp length product from the WT allele and a 370 bp product from the D47 allele. (**b**) Representative image of amplicons from semi-nested PCR visualized on agarose gel. The upper and lower band correspond to WT and D47 allele, respectively. Full-length agarose gel is included in the Appendix A.

**Figure 3 biomolecules-09-00136-f003:**
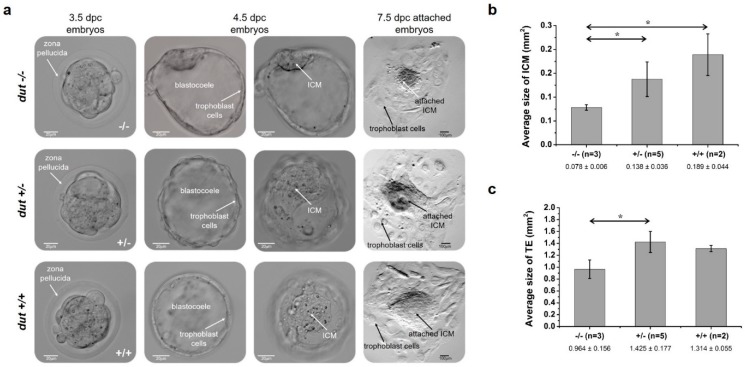
Outgrowth assay of pre-implantation embryos obtained by intercrossing D47 heterozygous mice. (**a**) Phase contrast images of D47 homozygous (−/−), heterozygous (+/–), and wild-type (+/+) blastocysts in in vitro culture. The first column shows embryos at 3.5 dpc after flushing from oviducts. White arrows indicate the zona pellucida surrounding the embryos. The second and third columns show the attached embryos, one day later focusing on the trophoblast cells or the inner cell mass (ICM) in the blastocoel. Scale bar represents 20 µm. The last column presents outgrowths after four days in culture. Scale bar represents 100 µm. Average size of ICM (**b**) and trophectoderm (TE) (**c**) was calculated for blastocysts of indicated genotypes. Error bars indicate standard deviation. n = 3 for (−/−), n = 5 for (+/–), and n = 2 for (+/+). Statistical analysis was done with a two-sample, single-tailed t-test assuming equal variance using Microsoft Excel. * *p* < 0.05.

**Figure 4 biomolecules-09-00136-f004:**
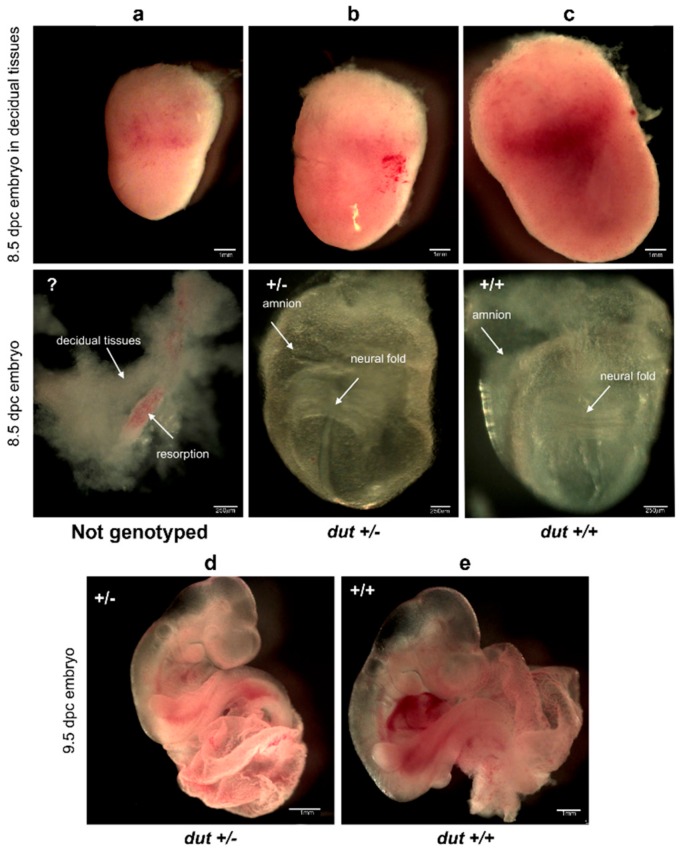
Images of embryos at 8.5 and 9.5 dpc obtained by crossing D47 heterozygous mice. (**a**) Representative image of a resorbed embryo at 8.5 dpc covered by decidual tissues. The resorbed embryo could not be genotyped as indicated with “?”. Heterozygous (+/–) (**b**) and wild-type (+/+) (**c**) embryos at 8.5 dpc are shown. Upper panels show embryos in intact decidual tissues. Scale bar represents 1 mm. Lower panels show the embryos dissected from decidual tissues. Arrows indicate the embryonic neural fold and the extra-embryonic amnion. Scale bar represents 250 µm. Heterozygous (+/–) (**d**) and wild-type (+/+) (**e**) embryos at 9.5 dpc are also shown. Scale bar represents 1 mm.

**Figure 5 biomolecules-09-00136-f005:**
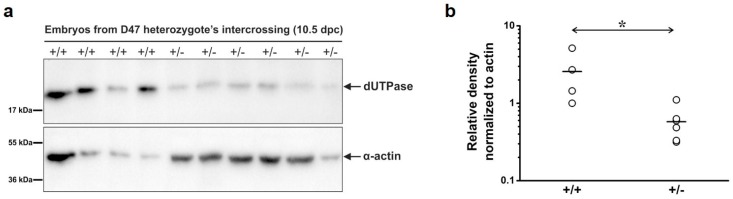
Protein level of dUTPase in embryos at 10.5 dpc from the intercrossing of D47 heterozygous mice. (**a**) Western blot displaying dUTPase protein level in wild-type (+/+) and heterozygous (+/–) mice. Membrane was developed against dUTPase (upper part) or α-actin (lower part) as a loading control. Blots are marked with black frame and separated with space. Uncut scans are shown in Appendix A. (**b**) Densitometric data for dUTPase levels from Western blot normalized for α-actin. Mean values are represented with horizontal lines. Every data point is shown, n = 4 for (+/+) and n = 6 for (+/–). Statistical analysis was carried out with a two-sided Mann–Whitney U test. * *p* < 0.05.

**Figure 6 biomolecules-09-00136-f006:**
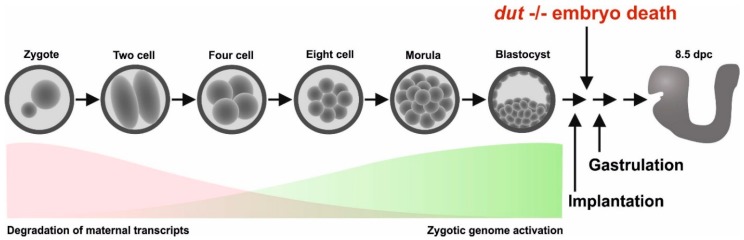
Schematic diagram of embryonic development in mice. The red and green areas show the timeline of simultaneous degradation of maternal transcripts and activation of zygotic transcription. Arrows illustrate that homozygous knock-out embryos die shortly after implantation.

**Table 1 biomolecules-09-00136-t001:** Potential off-target sites predicted by CCTop—CRISPR/Cas9 target online predictor software.

Name	Coordinates	MM	Target Sequence	PAM
**Dut**	**chr2:125247853-125247874**	**0**	**CGCGCGCGGAC[CCGCGGGT]**	**GGG**
**Off-1**	**chr13:88680616-88680637**	**4**	**TGAGTGGGGAC[CCGCGGGT]**	**TGG**
Off-2	chr4:120746882-120746903	4	CTGGAGCGGCC[CCGCGGGT]	GGG
**Off-3**	**chr15:28025084-28025105**	**4**	**CGGCCACGGCC[CCGCGGGT]**	**AGG**
Off-4	chr11:23306808-23306829	4	GGGGCGGGGAG[CCGCGGGT]	GGG
**Off-5**	**chr2:118372992-118373013**	**4**	**CGCTGGTGGCC[CCGCGGGT]**	**TGG**
**Off-6**	**chr5:64803646-64803667**	**4**	**ACCGCACGGAC[GCGCGGGT]**	GGG
**Off-7**	**chr18:85179644-85179665**	**3**	**CGTGCGCGCAC[GCGCGGGT]**	**GGG**
**Off-8**	**chr9:77319229-77319250**	**4**	**CGCGCTTACAC[CCGCGGGT]**	**GGG**
**Off-9**	**chr2:104319532-104319553**	**3**	**CGCGTGCGCAC[ACGCGGGT]**	**AGG**
**Off-10**	**chr19:36918725-36918746**	**4**	**CTCGCTGGGAC[GCGCGGGT]**	**AGG**
Off-11	chr5:75044665-75044686	4	TGGGCGCGGGC[GCGCGGGT]	GGG
Off-12	chr4:152086570-152086591	4	CGCACCCAGAC[ACGCGGGT]	CGG
Off-13	chr3:41563582-41563603	4	GGCGCGGGGGC[GCGCGGGT]	CGG
Off-14	chr8:60640130-60640151	4	GGCGCGTGGGC[ACGCGGGT]	TGG
Off-15	chr9:40192333-40192354	3	CGCGCGGGGCC[CAGCGGGT]	CGG
**Off-16**	**chr2:174438958-174438979**	**4**	**AGCGCGTGGGC[CTGCGGGT]**	**CGG**
Off-17	chr17:88792070-88792091	4	CGGGCGGGGGC[CGGCGGGT]	GGG
**Off-18**	**chr2:28641663-28641684**	**4**	**G** **GCACGGGGAC[CCGGGGGT]**	**GGG**
Off-19	chr17:28350853-28350874	4	GGCGGGCGGGC[CCACGGGT]	GGG
Off-20	chr5:107597539-107597560	4	GGCGCGTGGAT[CGGCGGGT]	AGG

The table presents the chromosomal location, the number of mismatches (MM), the target, and the adjacent PAM sequences. The first row depicts the target site of the designed gRNA, further rows list the top 20 candidates for off-target sites. Mismatches are indicated in red. Brackets include core sequences [34]. The genomic segments that were successfully sequenced are shown in bold.

**Table 2 biomolecules-09-00136-t002:** Genotype analysis of offspring from *dut* +/− intercrosses at different developmental stages.

DNA Source	Genotype			Resorbed	No. Total
	+/+	+/−	−/−		
Postnatal	21	42	0	NA ^a^	63
10.5 dpc	3	5	0	3	11
9.5 dpc	5	5	0	0	10
8.5 dpc	10	5	0	5	20
3.5 dpc	11	13	7	NA	31

^a^ NA, not applicable.

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
