# Peer review of "CRISPR/Cas9-Mediated Knock-Out of dUTPase in Mice Leads to Early Embryonic Lethality"

_biomolecules, 2019, doi:10.3390/biom9040136_

Round 1

Reviewer 1 Report

This study describes the generation of dut knock-out mice using the CRISPR/Cas9 methodology. The authors demonstrate that, similarly to other organisms, dUTPase deficiency is incompatible with normal life and, in mammals, with development and survival. 

In the Discussion section, the authors speculate on the reasons underlying the survival of blastocysts and posit that it might be linked to the remaining pool of maternal dUTPase. Because incorporation of dUMP into DNA is not mutagenic, the DNA may perform its normal function so long as the diverse uracil glycosylases (UDG, TDG, SMUG1, MBD4) do not begin to remove the uracil. Once this happens, the DNA will acquire numerous breaks that will eventually lead to cell death. It might be interesting to discuss the temporal appearance of uracil-processing enzymes in relation to dUTPase.

It would be interesting to see the phenotype of the dut/ung double KO mice in future studies.

The manuscript is well written, but the authors ought to change the tense of the Results section (and some sentences in the Discussion). In general, experimental findings are described in the past tense.

Author Response

Thank you for your expert review.

We fully agree that a discussion on the temporal appearance of uracil-processing enzymes in relation to dUTPase would be highly relevant. Unfortunately, as we mentioned in the Discussion, reliable data on the expression levels of uracil processing enzyme along mouse embryonic development are not yet published. We also agree that the phenotype of dut/ung double KO mice would be highly interesting and we working forward to construct these transgenic animals.

As you proposed, we changed the Results and Discussion sections to past tense.

Reviewer 2 Report

Overall, the scientific content of the manuscript is solid and important. The experimental design is appropriate and the conclusions are supported by their data.  My main concerns are the numerous grammatical errors/grammatically incorrect sentences and paragraphs throughout the manuscript. This must be corrected. For example:

Pg2 Line 63: Not grammatically correct sentence – “Bacterial cells are not viable lacking dUTPase activity” It should be rephrased to read “Bacterial cells lacking dUTPase activity are not viable”

Pg3 line 91: … PAM .. . Spell out the first time is stated in the manuscript

Pg3 line 99 ; “.. cells were maintained in a fresh medium”  delete “a” to read - in fresh medium

Pg3 line 102 .  … according to manufacturer’s instructions.  Insert “the” before manufacture’s

Pg3 line 103. dUTPase “wild type” should have a hyphen between the two words to read wild-type

Pg3 line 116. Spell out hCG injection

 Pg4 Line 131. It should read “subject to sequencing”

Pg4 line 141. “aspecific”PCR . Aspecific, is not a commonly used term to refer to a non-specific PCR product and would suggest to change it to simple “non-specific”

Pg4 line 159  Insert “a” before LeicaM205FCA-FA ….

Section 2.10 Western Blot. State what homogenization method/equipment was used

Pg 8 line 258 Delete (days post coitum). It has already been defined earlier in the text

Under Section 3.3 Embryonic  Development …. Pg 9 Line 284-86: Quantitative analysis of these images was also initiated”….  It should be re-phrased and state the specific areas that are being analyzed. For ex: “Quantitative analysis of ICM and TE regions were performed as described in supplementary figure 4.

Pg10 lines 304-305. The first sentence is not grammatically correct and should re-stated

In supplementary figure S5 legend.  (1) The term “Neural groove” is not used in the pictures/images.  Instead the term Neural growth is used. Stay consistent with whatever terminology is used. (2) When possible the embryonic regions and extra-embryonic tissues should be labeled in all the pictures/images. (3) Images were taken at different magnifications. For consistency, all images should be at the same magnification

Author Response

Thank you for your expert review.

We agree to all of your remarks and thank you for the suggested corrections.

Accordingly, we have corrected the revised version of our manuscript. 

We also thank you for the suggested improvement of Supplementary Figure S5. We agree to your suggestion and have modified this figure in the revised version.

Reviewer 3 Report

Overall this is an excellent manuscript. The subject matter is important and the manuscript is well written

I thought it was an excellent manuscript. I could not find any flaws in the technical aspects of the research, the writing was very good and the conclusions were reflective of the study. Furthermore, the study addressed a question that needed to be answered. Overall in my opinion the manuscript is acceptable for publication as it stands.

Author Response

Thank you very much for your positive opinion on our manuscript. It was a pleasure to learn that our study is acceptable for publication.